# Potential Impact of Meat Replacers on Nutrient Quality and Greenhouse Gas Emissions of Diets in Four European Countries

**Elly Mertens** [1,2], **Sander Biesbroek** [1,*], **Marcela Dofková** [3], **Lorenza Mistura** [4], **Laura D'Addezio** [4], **Aida Turrini** [4], **Carine Dubuisson** [5], **Sabrina Havard** [5], **Ellen Trolle** [6], **Johanna M. Geleijnse** [1,2] and **Pieter van 't Veer** [1,2]

1. Division of Human Nutrition and Health, Wageningen University, P.O. Box 17, 6700 AA Wageningen, The Netherlands; elly.mertens@wur.nl (E.M.); marianne.geleijnse@wur.nl (J.M.G.); pieter.vantveer@wur.nl (P.v.V.)
2. TiFN, P.O. Box 557, 6700 AN Wageningen, The Netherlands
3. Center for Health, Nutrition and Food, National Institute of Public Health, 61242 Brno, Czech Republic; dofkova@chpr.szu.cz
4. Council for Agricultural Research and Economics, Research Centre for Food and Nutrition, Via Ardeatina 546, 00178 Rome, Italy; lorenza.mistura@crea.gov.it (L.M.); laura.daddezio@crea.gov.it (L.D.); aida.turrini@crea.gov.it (A.T.)
5. French Agency for Food, Environmental and Occupational Health & Safety (ANSES)/Risk Assessment Department (DER, 14 rue Pierre et Marie Curie, 94701 Maisons-Alfort Cedex, France; carine.dubuisson@anses.fr (C.D.); sabrina.havard@anses.fr (S.H.)
6. Division of Risk Assessment and Nutrition, National Food Institute, Technical University of Denmark, Kemitorvet Bygning 202, 2800 Kongens Lyngby, Denmark; eltr@food.dtu.dk
* Correspondence: sander.biesbroek@wur.nl

**Abstract:** Meat replacers could play a role in achieving more plant-based diets, but their current consumption is limited. The present modelling study aimed to explore the nutritional and greenhouse gas emissions impacts of meat replacers. Using dietary surveys from Denmark, Czech Republic, Italy and France (~6500 adults), we composed alternative diets in which all the meat in the observed diet (in grams) was substituted by similar use meat replacers (with and without fortification). Starting from the observed diets and meat-replacement diets, diets with improved adherence to food-based dietary guidelines (FBDGs) were modelled using Data Envelopment Analysis. These improved diets were then further optimised for dietary preferences (MaxP, diet similarity index), nutrient quality (MaxH, Nutrient Rich Diet score, NRD15.3) or diet-related greenhouse gas emissions (GHGE) (MaxS, $CO_2$ equivalents). In all optimised modelled diets, the total amount of meat was lower than in the observed diets, i.e., 30% lower in the MaxP, 50% lower in the MaxH, and 75% lower in the MaxS diets. In the MaxP diet, NRD15.3 was ~6% higher, GHGE was ~9% lower, and ~83% of food intake remained similar. In the MaxH diet, NRD15.3 was ~17% higher, GHGE was ~15% lower, and ~66% of food intake remained similar. In the MaxS diet, NRD15.3 was ~9% higher, GHGE was ~33% lower, and ~65% of food intake remained similar. When using fortified meat replacers, for all modelled diets, the diet similarity was on average 2% lower and the GHGE reduction was on average 3% higher as compared with the same scenarios without fortification. This analysis showed that meat replacers, provided their preference is similar to meat, can provide benefits for GHGE, without necessarily compromising nutrient quality.

**Keywords:** diet modelling; environmental impact; greenhouse gas emissions; nutritional quality; preferable; scenario analysis

## 1. Introduction

Healthy and environmentally sustainable diets are key to the Sustainable Development Goals as well as the Paris Climate agreement [1,2]. Currently, food systems are responsible for a significant share (20–33%) of greenhouse gas emissions (GHGE) and land use change, and a major driver of deforestation and loss of biodiversity [1]. With the world's population predicted to expand to around 10 billion by 2050, change is needed [1,2].

Since the introduction of the report of the EAT-Lancet Commission, much attention has been paid to its implications for current dietary habits across the world [3]. In general, such healthy diets have an adequate caloric intake and consist of high consumption of plant-based foods, low amounts of animal-based foods, unsaturated rather than saturated fatty acids, and small amounts of refined grains, highly processed foods, and added sugars [3]. The difference with current dietary habits is large and requires major changes in agriculture, diets and policies and industrial practices, therefore this process is also called the 'Great Food Transformation'.

In Western countries, increasingly more varieties of meat replacers are available on the market [4,5]. With total fat and saturated fatty acids often lower, salt content higher, and total protein lower the nutritional composition of meat replacers is not necessarily healthier than meat [4,6]. These meat replacers are often designed to resemble the look, taste, and structure of meat. Smetana et al. showed that, with current technology, the highest environmental impacts are for lab-grown meat and mycoprotein-based analogues, medium impacts for dairy- and gluten-based meat substitutes, and the lowest impacts for insect- and soy-based substitutes [7]. The environmental impact profiles of the meat replacers are lower in GHGE, water, and land use compared to the animal-sourced food that they aim to replace [8]. Nevertheless, the values of environmental impact would be subject to estimate variability/uncertainty related to, but not limited to, the country of agricultural production, country of processing, and the inclusion of additional nutrients for fortification.

In a previous study, we used Data Envelopment Analysis (DEA) to identify diets that score relatively high on criteria of food-based dietary guidelines (FBDGs), and subsequently use these diets to model improved diets [9,10]. In this way, by the use of peer resemblance, improved diets are still within the boundaries of observed dietary practices, and hereby representing improvements in the proper direction rather than ultimate goals for dietary improvement. Nevertheless, on average, such improved diets showed 6–16% higher Nutrient Rich Diet scores (NRD15.3) as well as 3–21% lower GHGE. As industrially produced meat replacers were virtually absent in the existing dietary practices in these countries before 2010, they were not included in these models. Therefore, in the current study we included meat replacers as a potential substitute and studied how this affects the deviation from current diets, GHGE, and nutrient quality of the modelled diets. Moreover, we evaluated how results were affected by theoretical fortification of existing meat replacers with iron, and vitamins B1, B2, B3, and B12 up to 30% of the Nutrient Reference Values. The results of the present study will rely on the assumption that individuals are willing to substitute their meat by meat replacers on an equal weight basis.Peer resemblance thus assumes consumers are willing to adopt the diet of others. This could provide improved diets within the boundaries of currently observed dietary practices—thus improvement in the proper direction, which can be gradually progressed—without yet having ultimate goals for health and environmental sustainability.

## 2. Materials and Methods

### 2.1. Study Population and Observed Dietary Data

Individual-level dietary data used in the present study were collected by nationally representative dietary surveys in four European countries, i.e., DANSDA (2005–2008) in Denmark (DK), based on seven-day diet records on consecutive days [11]; SISP04 (2003–2004) in the Czech Republic (CZ), based on two 24-h recalls spaced over three to five months [12]; INRAN-SCAI (2005–2006) in Italy (IT), based on three day–day diet records on consecutive days [13]; and INCA-2 Study (2006–2007) in France

(FR), based on seven-day diet records on consecutive days [14]. Included for each country were two non-consecutive days for the adult population, aged 18–64 years. After the exclusion of mis-reporters using the Goldberg equation [15] as adopted by Black [16], present analyses were conducted on a final sample of 1385 adults in Denmark, 1386 adults in the Czech Republic, 1978 adults in Italy, and 1713 adults in France.

All the foods reported were classified for each country according to the FoodEx2 classification developed by the European Food Safety Authority (EFSA) [17,18]. Intakes of foods were calculated from the mean of two days for each individual, and were expressed per 2500 kcal for men and per 2000 kcal for women. Nutrient intakes were calculated using country-specific food composition databases [19–25]. The GHGE (in kgCO$_2$ equivalents (kgCO$_2$ eq)/kg food as eaten) were calculated using a standardised life cycle assessment (LCA) database of GHGE for the 944 FoodEx2 codes as consumed in the four countries (SHARP-Indicators Database (SHARP-ID) [26]). In this database, the life cycle inventory data of 182 primary products were combined with data on production, trade and transport, and adjusted for consumption amount using conversions factors for production, edible portion, cooking losses and gains, and for food losses and waste in order to derive estimates of GHGE for the foods as eaten. GHGE was expressed in kilogram CO$_2$ equivalents (kgCO$_2$ eq) per kg primary product, with 1 kgCH$_4$ equal to 25 kgCO$_2$, and 1 kgN$_2$ O equal to 298 kgCO$_2$ (IPCC 2007). General aspects of dietary choices and the associated environmental impact in these four European countries have been published by Mertens et al. [27].

## 2.2. Meat-Replacement Diets

To be able to introduce meat replacers to the model, we enriched the dataset with newly constructed diets. We essentially doubled the observed food intake data, and subsequently substituted in each duplicated diet 100% of the reported meat consumption with the same quantity of meat replacers (in grams), without altering the consumption of any of the other foods. Therefore, at the start of the analysis we had a dataset with two dietary observations per person—one being the observed diet and the other the diet with substituted meat by meat replacers.

Available meat replacers and their nutrient content, as reported in the Dutch Food Composition Database 2016, were included [28]. Meat products were replaced by similarly used meat replacers, for example a hamburger was replaced by a vegetarian hamburger and a sausage by a vegetarian sausage to stay as close as possible to the observed diet (Supplementary Table S1). Values of GHGE were obtained from Blonk Consultants [8], of which the LCA methods are comparable to our SHARP-ID [26]. Simple substitution was used to re-calculate the nutrient composition and GHGE of the meat-replacement diets after standardization for energy intake (2500 kcal for men and 2000 kcal for women). Next to this, to evaluate the potential influence of fortification, the meat replacers were theoretically fortified with iron (4.2 mg/100 g), vitamin B1 (0.33 mg/100 g), vitamin B2 (0.42 mg/100 g), vitamin B3 (4.8 mg/100 g) and vitamin B12 (0.75 μg/100 g), corresponding to the 30% of the Nutrient Reference Values (NRVs) [29]. As the fortification with iron, vitamin B12 or other micronutrients adds less than 1% on a weight basis, for these fortified foods the same GHGE was used as the original products.

In the diet modelling, three diets were used, each based on the average of two non-consecutive days and expressed per 2500 kcal for men and per 2000 kcal for women. The first diet was the observed diet. The second diet was the meat-replacement diet in which for each individual all the amounts of the meat consumed in the observed diet was replaced by a sensible meat replacer of the same quantity (in grams). The third diet was the fortified meat-replacement diet using meat replacers with fortification levels as indicated above instead. In our dataset, less than 5% was a non-meat consumer, and for them both meat-replacement diets were identical to the observed diet.

### 2.3. The Benchmarking Diet Model

Data Envelopment Analysis (DEA) was used to provide dietary improvement options that implicitly account for dietary preferences, as described by Kanellopoulos et al. [30] and Mertens et al. [9,10]. Briefly, starting from a set of diets, the DEA model was used to identify efficient diets, with a higher benefit/cost ratio, i.e., diets that have a higher ratio of "dietary components to increase" (benefits) to "dietary components to decrease" (costs). Subsequently, linear combinations of the efficient diets were used to optimise for either preferences (minimal deviation from observed diet), health (nutrient score) or environmental sustainability (GHGE). The DEA-model was run using FICO Xpress-IVE version 1.25.02. Diets were modelled for each country, and for men and women separately.

### 2.4. Identification of Efficient Diets

The benchmarking started from a set of diets that included for all individuals both their observed diet and their meat-replacement diet, and from this set of diets efficient diets were identified. Similar to the study of Mertens et al. [9,10], the identification of efficient diets was primarily based on FBDGs and consisted of eleven dietary components to increase, i.e., fruit, vegetables, legumes, nuts and seeds, fish, whole grains, unsaturated fatty acids, calcium, zinc, vitamin B2 and vitamin B12, and five dietary components to decrease, i.e., red and processed meat, sweet beverages, refined grains, ethanol and saturated fatty acids. With mostly foods and the fatty acids included in the model based on the FBDGs, we observed unwanted changes in some nutrients. Therefore, calcium, zinc, vitamin B2 and vitamin B12 were added to the model to be safeguarded. To evaluate the potential role of fortification, the benchmarking was subsequently repeated for a set of diets that included both the observed diet and the fortified meat-replacement diet. Supplementary Table S2 shows the diets that were labelled as efficient.

### 2.5. Modelled Diets

Three different options for dietary improvement were explored, i.e., modelled diets that remain as close as possible to the observed diet (MaxP), have the highest nutrient quality (MaxH), or have the lowest GHGE (MaxS), as described in more detail elsewhere [9,10].

To model the MaxP diets, we maximised dietary preferences by minimising the sum of the positive and negative deviations (in absolute values) of food group intake from the observed diet, i.e., the minimum deviation (MINDV) approach [30]. To simply describe the dietary preferences, we used the diet similarity index, i.e., the summed amount of each food group that remained the same in the modelled diet as compared to the observed diet divided by total diet weight of the observed diet [9,10]. To model the MaxH diets, we maximised the NRD15.3 score [31,32], i.e., the unweighted sum of percentage daily values for 15 nutrients to encourage (protein, mono-unsaturated fatty acids, dietary fibre, calcium, iron, potassium, zinc, vitamin A, D, E, C, B1, B2, and folate) minus the sum of percentage maximum recommended values for three nutrients to limit (saturated fatty acids, added sugar, and sodium), calculated per 2500 kcal for men and 2000 kcal for women, and capped at 100% of the dietary recommended intake. To model the MaxS diets, we maximised environmental sustainability by minimising GHGE. Results of the diet similarity index, NRD15.3 and the GHGE were expressed relative to the observed diets; in addition, the food and nutrient composition of modelled and observed diets were compared. The population prevalence of inadequate intake of nutrients for the different diets is estimated by means of the EAR cut-point method without correction for within subject variability and using the Population Reference Intake (PRI) values estimated by EFSA if available [33]. If not available, the Adequate Intakes (AI) in the same report were used.

## 3. Results

### 3.1. Dietary Preferences, Nutrient Quality and Environmental Impact of the Modelled Diets

Figure 1 shows the evaluation of the nutrient quality and the GHGE of the modelled diets. When including meat replacers in the modelled diets, the MaxP diets had a 4–9% higher NRD15.3 score, but also had 4–12% lower GHGE than the observed country-gender-specific diets. The MaxP diets remained closest to the observed diet (diet similarity index on average 83%; results not shown). For the MaxH and the MaxS diets, further improvements in nutrient quality and environmental impact occurred at the expenses of dietary preferences (similarity index on average 65% for both—results not shown). For the MaxH diets, the NRD15.3 was 11–24% higher, and the GHGE was 9–21% lower than observed. Likewise, for the MaxS diets, the GHGE was 24–39% lower than observed, and the NRD15.3 was 4–15% higher than observed. Moreover, when using fortified meat replacers, for all modelled diets, the diet similarity was on average 2% lower and the GHGE was on average 3% lower as compared with the same scenarios without fortification, while the average NRD15.3 score of the modelled diets was not influenced by the fortification.

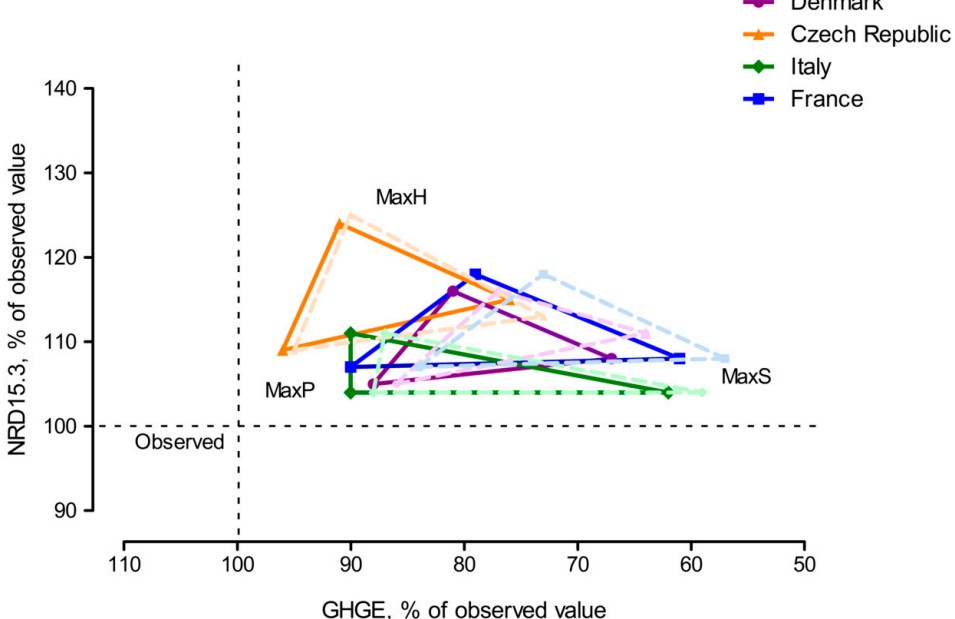

**Figure 1.** Evaluation of the nutrient quality and the greenhouse gas emissions of the modelled diets according to a benchmarking diet model and using meat replacers [a,b]. ([a] All modelled diets have an improved adherence to food-based dietary guidelines, MaxP is the most preferred diet based on minimal deviation from the observed diet, MaxH the most healthy diet based on NRD15.3 for nutrient quality, and MaxS the most environmentally sustainable diets based on GHGE. [b] Meat replacers as available on the market represented by the dark coloured lines and fortified meat replacers by the light-coloured dashed lines).

### 3.2. Meat Consumption in Observed and Modelled Diets

Figure 2 and Supplementary Table S3 present the diet composition of the main food groups in the observed and the modelled diets based on the unfortified meat replacers. In the observed diets, meat replacers were not present, while they were included in all the modelled diets; in most population subgroups, their amount showed an upwards gradient over the scenarios of MaxP, MaxH and MaxS. The amount of meat in the observed diets ranged from 113 (IT) to 175 (FR) gram per 2500 kcal for men and from 85 (DK) to 122 (FR) gram per 2000 kcal for women. In all modelled diets, for both men and women, the amount of meat was lower than in the observed diets, in particular around 30% lower in the MaxP, 50% lower in the MaxH, and 75% lower in the MaxS diets. Moreover, when using fortified

meat replacers, the amount of meat in the scenarios of MaxP, MaxH and MaxS was lower than in the same scenarios without fortification, i.e., on average 10 g lower for men and 6 g lower for women in all corresponding modelled diets (Supplemental Table S3). Thus, the lowest modelled amount of meat was seen in the MaxS diets with fortified meat replacers, with amounts ranging from 19 (IT) to 39 (CZ) gram per 2500 kcal for men and from 6 (FR) to 27 (CZ) gram per 2000 kcal for women. The highest modelled amount of meat replacers was seen for men in the MaxS diets with fortified meat replacers with amounts ranging from 101 (IT) to 140 (FR) grams per 2500 kcal, and for women in the MaxH diets with fortified meat replacers in Denmark (61 g per 2000 kcal) and France (173 g), but not in the MaxS diets with fortified meat replacers in Czech Republic (70 g) and Italy (76 g).

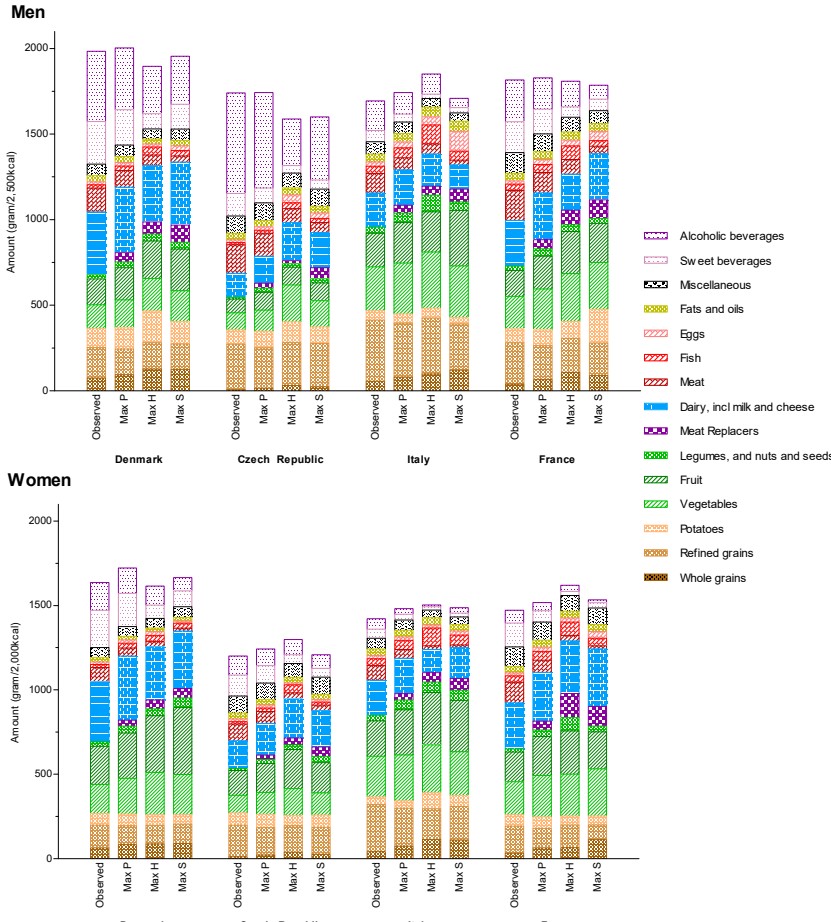

**Figure 2.** Mean quantities for main food groups in the observed and modelled diets (based on unfortified meat replacers) [a,b,c]. ([a] All modelled diets have an improved adherence to food-based dietary guidelines, MaxP is the most preferred diet based on minimal deviation from the observed diet, MaxH the most healthy diet based on NRD15.3 for nutrient quality, and MaxS the most environmentally sustainable diets based on GHGE. [b] Information on amount of coffee, tea, and water in the diets is not included in the graph. [c] Average amount of food consumed by the population in gram per 2500 kcal for men and per 2000 kcal for women).

### *3.3. Overall Dietary Pattern in Observed and Modelled Diets*

When looking at the overall dietary pattern, in all countries, and for both genders, the proportion of animal-sourced foods was lower in the modelled diets than in the observed diets, in particular around 5% lower in the MaxP diets and around 10% lower in the MaxH and the MaxS diets (Figure 2). This lower proportion of animal-sourced foods was mainly attributed to the lower amounts of meat, as mentioned in the previous section. In the MaxS diets, this proportion of animal-sourced foods was in most population subgroups further lowered when using fortified meat replacers as compared with

diets without fortification, in particular up to 5% lower in men of Denmark and the Czech Republic, and in women of Denmark, and up to 9% lower in women of France. This was mainly due to the lower amounts of dairy when using fortified meat replacers compared with no fortification. Nevertheless, irrespective of the fortification, the amount of animal-sourced foods was higher than observed in the MaxH diets of men in the Czech Republic (+15%) and Italy (+6%) and of women in the Czech Republic (+11%). This higher amount could be attributed to a higher amount of fish, and for the Czech Republic also to a higher amount of dairy and eggs (men only). In particular, the amount of fish was more than doubled in all the MaxH diets for all countries and for both genders (Supplementary Table S3). Total amount of dairy was higher in the modelled diets of men in the Czech Republic and women in the Czech Republic and France. For both countries, the highest amount of dairy was included in the MaxH and MaxS diets (+13 to +60%).

In all countries, and for both genders, the total amount of plant-sourced foods was higher in the modelled diets than in the observed diets (+13% to +53%), while the total amount of sweet and alcoholic beverages was lower (−78% to −10%) (Figure 2). In addition to the inclusion of meat replacers, as mentioned above, the amounts of fruits, vegetables, legumes and whole grains were higher, whereas the amount of refined grains was lower, and this was the most pronounced for the MaxH and the MaxS diets. The amount of nuts and seeds was only slightly higher than in the observed diet. The total amount of sweet and alcoholic beverages was the lowest in the MaxS diets, while the amount of sweet beverages was the lowest in the MaxH diets (Supplemental Table S3). When using fortified meat replacers, the total amount of plant-sourced foods in the scenarios of MaxP, MaxH and MaxS was higher than in the same scenarios without fortification for most population subgroups (+1% to +19%), while the amount of sweet and alcoholic beverages was not materially affected more (Supplemental Table S3).

*3.4. Nutrient Quality of Observed and Modelled Diets*

Supplementary Table S4 shows the estimated prevalence (%) of nutrient inadequacies for the observed and the modelled diets. In our data, the three scenarios of MaxP, MaxH and MaxS resulted in improved nutrient adequacies of dietary fibre, magnesium, calcium, potassium, vitamin E, vitamin B1, vitamin B2, vitamin B12 and folate. However, the prevalence of vitamin B3 inadequacy was higher in most modelled diets as compared to the observed diets, except for Denmark.

When comparing the scenarios of MaxP, MaxH and MaxS, a higher prevalence of inadequacies was observed in the MaxP diets, with more than 50% of the population having an inadequate intake in all countries for vitamin D, for iron (in women only), and for dietary fibre and potassium (except for Danish men and women, and Italian men). Although the average vitamin B12 intake was lower than in the observed diets, the prevalence of inadequacies decreased in some of the scenarios, since especially low meat consuming people increased their vitamin B12 intake by consuming meat replacers, more fish and dairy.

When using fortified meat replacers, the scenarios of MaxP, MaxH and MaxS resulted in less inadequate intakes than in the same scenarios without fortification, in particular, for example, for the MaxS diets the fortified nutrient vitamin B2 in men and women of the Czech Republic (63% vs. 43% for men, and 67% vs. 55% for women). Similarly, for example, for the MaxS diets, when using fortification, the prevalence of vitamin B3 inadequacy was lower in men of the Czech Republic (28% vs. 11%), Italy (24 vs. 12%) and France (43% vs. 17%), and in women of the Czech Republic (67% vs. 46%), Italy (41% vs. 21%), and France (55% vs. 26%). Scenarios with fortified meat replacers did, however, not consistently result in less inadequate intakes for vitamin B12 than in the same scenarios without fortification.

## 4. Discussion

In this modelling study, the benchmarking started from a set of diets that included for all individuals both their observed diet and their meat-replacement diet to identify diets that scored relatively high

on criteria of FBDGs. Linear combinations of these diets were then optimized for similarity (MaxP), nutrient quality (MaxH), or GHGE (MaxS). In all optimised modelled diets, the average total amount of meat of the four countries was lower than in the observed diets, i.e., 30% lower in the MaxP, 50% lower in the MaxH, and 75% lower in the MaxS diets. The modelled diets that were the closest to the observed diets (MaxP) had on average a 6% higher nutrient quality and on average an 8% lower diet-related GHGE. The nutrient quality could be further improved up to 17% in the MaxH scenario, and the diet-related GHGE could be reduced up to 39% in the MaxS scenario. When using fortified meat replacers, further improvements in diet-related GHGE were observed on average over the four countries, in particular up to 42% in the MaxS scenario, without further improving the nutrient quality of the modelled diets. Although results differed by country and by gender, all modelled diets had a higher amount of plant-sourced foods, with on average a 5% lower proportion of animal-sourced foods in the MaxP diets, and on average a 10% lower proportion in MaxH and MaxS diets. When using fortified meat replacers, the amounts of meat replacers were higher and the amounts of meat were lower, with in the MaxS diets also lower amounts of dairy. Interestingly, even though meat consumption decreased in the MaxS scenarios vitamin B12 inadequacies decreased in some scenarios, since people that originally consumed no meat or small amounts of meat could now include the vitamin B12-rich meat replacers in their recommended diets.

In accordance with the present results, previous studies have demonstrated that replacing meat is a cornerstone for reducing diet-related environmental impact [34,35]. In the context of meat replacers, decisive factors for successful adoption of them are taste and appearance for repeated consumption, but also involve easy availability and compatibility with local foods [36]. To stay close to the observed dietary practices, the meat replacers available on the market are of similar intended use. In this way, meat replacers are seen as an easy-to-use option for replacing meat due to their convenience and their resemblance to meat allowing to maintain the component structure of the meal [37]. In addition, with the option of meat replacers, the minimum reachable GHGE was 10 percentage points lower as compared to the same scenarios without meat replacers, as done in our previous analyses [9,10]. This environmental benefit was neutral for NRD15.3, but occurred at the expense of dietary preferences, i.e., diet similarity between observed and modelled diets was 5 percentage points lower compared to our previous analyses [9,10]. Nevertheless, under the assumption that meat replacers may replicate the sensory experience of eating meat, results of this study highlight that meat replacers could support a shift towards more plant-based diets. It also shows that other changes in food consumption are needed to safeguard a nutrient-rich diet. Our modelled diets have a larger proportion of plant-based foods, while in our previous analyses without meat replacers this proportion remained unchanged [9,10]. In addition to the commercial meat replacers, a more plant-based diet generally emphasises higher amounts of fruit, vegetables, whole grains, legumes and nuts and seeds, which is in line with the food-based dietary guidelines.

Our modelling study only considered replacing meat by meat replacers, while according to EAT-Lancet Commission [3], a reference diet should include higher amounts of the plant-sourced protein-rich food groups, such as legumes, nuts and seeds. Such a diet was projected to lower GHGE by around 50% in 2050 as compared to the current diets [3]. However, using legumes, nuts and seeds as replacement foods modifies the component structure of the meal and might therefore impede diet acceptability. On the other hand, other consumer segments might look for meat-free meals that are different from conventional ones, hence not using meat-like replacement foods. With the use of the DEA methodology of introducing improved diets to the model, it is possible to model small steps in the transition towards healthy and environmentally sustainable diets. This modelling study explores the first steps of moving beyond the current dietary practices by introducing meat replacers to the model.

Replacing all meat by a (fortified) meat replacer can lead to a 20–30% lower GHGE than observed (Supplemental Table S5). However, consuming only these meat replacers instead of meat is not a guarantee of a healthier diet. In such full replacement diets, estimated amounts of sodium were higher, and prevalence of inadequacies of zinc, vitamins A, B1, B3 and B12 were higher than

observed (Supplemental Table S6). In another modelling study, it was shown that it is theoretically possible to compose meat replacers with ingredients like legumes, wheat, eggs and dairy, that have a meat-equivalent nutrient profile for energy, iron, zinc, vitamin B12 and the essential amino acids, with a lower environmental impact [38]. However, the use of additional fortifications was needed to compose a nutritionally equivalent product to beef due to the amount of iron, vitamin B12 and zinc present in beef [38].

Applying the DEA diet model for the design of improved diets has the advantage that dietary preferences are implicitly accounted for. In particular, because the model uses linear combinations of diets rather than foods, there is no need to introduce additional constraints on intrinsic relations between food products, such as bread and butter, cereals and milk. However, the use of peer resemblance allows only options for dietary improvement within the boundaries of observed dietary practices. Since our current diets are far from the FBDGs, our modelled diets provide more likely and realistic improvements into the proper direction, which can be gradually progressed, and not yet the ultimate goals for healthy and environmentally sustainable diet such as the reference diet of the EAT-*Lancet* commission [3], which on average is only 40% similar to the current diets, hence they would require considerably more changes. Mertens compared the EAT-*Lancet* diet to the benchmarking diets and observed diets of the four countries in her PhD thesis [10]. The reduction in GHGE of the EAT-*Lancet* diet would be on average 40%, while the average reduction in our meat replacer case study was around 30%.

Some limitations of the present study are related to the dietary surveys. In particular, the used data were derived between 2003 and 2008 (varying per country) and todays' dietary patterns and available foods are likely different than 15 years ago. The results of the analyses also depend on the validity of the reported diets, and data comparability between the countries was challenging, as discusses in our previous analyses [9]. Relying on two days of dietary recalls without adjusting for within-subject variation is likely to overestimate the prevalence of nutrient inadequacies when mean nutrient intake is higher than the DRV, and vice versa, but provided insight in the direction of improvement. Questions might also arise on the assumptions at the basis of the benchmarking diet model, such as the choice of the dietary components to identify efficient diets, as discussed in our previous analyses [9]. In addition, the use of peer resemblance for modelling improved diets within the ranges of observed dietary practices assumes that consumers are willing to adopt the diet of others. The observed diets dated from the early 2000s where meat substitution options were not widely available, whereas the last decade has seen a growing trend towards meat replacers. For the purpose of this modelling study, we hypothesised that consumers are willing to substitute meat by meat replacers and do that based on equal food weights. Such a replacement on weight basis, however, also slightly changed the total energy intake of the diet with the change of direction depending on the type of meat being replaced. On average, the amount of meat per energy percentage was somewhat higher than the amount of meat replacer per energy in Denmark and Italy, and somewhat lower for Czech Republic and France. After energy standardisation, amounts per kcal slightly differed between observed and meat-replacement diets, while the underlying basic interrelationship between food groups remained intact. Other issues that were not addressed in this study, because of limited data availability, were the protein quality (amino acid composition) of the meat replacers and the variability/uncertainty for the GHGE values of the meat replacers. Shifting from animal- to plant-based diets will also affect the protein quality of the diet, as the amino acid profiles of the latter are suboptimal for humans. Although relevant the necessary data of such profiles is limited or severely outdated. The GHGE values will be highly dependent on the country of agricultural production of the crops used for the meat replacers, as well as the country of processing due to, e.g., the different energy mix, transport distances, etc. Additionally, the production of the additional nutrients for fortification emits GHGs but, due to a lack of reliable data, this study was not able to quantify to what extent this would increase the GHGE of the fortified meat replacers. Due to the limited contribution on weight basis, we decided to use the impact of the original foods.

## 5. Conclusions

This modelling study showed the potential of meat replacers for the first steps towards healthier and lower GHGE diets under the assumption that they will become widely accepted by consumers. The results suggest that a partial shift from meat to meat replacers can lead to a one-third reduction in diet-related GHGE or up to a one-sixth improvement in NRD15.3, while keeping around 65% of the food group intake similar. In the modelled diets, (fortified) meat replacers (up to 30% of the nutrient reference values) alleviated the prevalence of some already prevalent nutrient inadequacies, even for vitamin B12. However, full replacement of meat increased the number of nutrient inadequacies. Therefore, additional dietary changes are required to arrive at healthy and environmentally sustainable diets, such as an increased consumption of vegetables, legumes, nuts and whole grains as suggested by the modelled diets.

**Supplementary Materials:** The following are available online at http://www.mdpi.com/2071-1050/12/17/6838/s1. Table S1: Meat replacers available from the Dutch Food Composition Table (NEVO) from 2016. Table S2: Results of the identification of efficient diets. Table S3: Amount of foods by food group for the observed and modelled diets. Table S4: Estimated prevalence of nutrient inadequacies for the observed and modelled diets. Table S5: Amount of foods by food group for the observed, full meat replaced, and full fortified meat replaced diets. Table S6: Estimated prevalence of nutrient inadequacies in the observed, full meat replaced, and full fortified meat replaced diets.

**Author Contributions:** Conceptualization, E.M., S.B., J.M.G. and P.v.V.; methodology, E.M. and S.B.; formal analysis, E.M.; writing—original draft preparation, E.M. and S.B.; writing—review and editing, M.D., L.M., L.D., A.T., C.D., S.H., E.T., J.M.G. and P.v.V.; visualization, E.M.; supervision, P.v.V.; All authors have read and agreed to the published version of the manuscript.

**Funding:** This research was funded by TiFN under Project Agreement number 15SD01 (SHARP-BASIC). E. Mertens, J.M. Geleijnse, and P. van 't Veer received research funding from TiFN (grant 15SD01_SHARP_BASIC). SHARP-BASIC was co-funded by the SUSFANS AF-EU-15027 project and supported by Unilever, DSM Nutritional Products, and the NZO (Dutch Dairy Association).

**Conflicts of Interest:** The authors declare no conflict of interest.

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
