# Peer review of "Potential Impact of Meat Replacers on Nutrient Quality and Greenhouse Gas Emissions of Diets in Four European Countries"

_sustainability, doi:10.3390/su12176838_

Round 1
Reviewer 1 Report
I agree for the changes made to the manuscript and related to other criticisms but I disagree on iron requirements. The AR (average requirements) is defined as the iron intake that cover the requirements of half the population, but this imply that the other half of the population has an inadequate supply of the mineral.
Authors should have used the PRI (population reference intake), which is defined as the nutrient intake that cover the requirement of the 97.5% of the population.
Using this approach the iron requirement would became 12 mg/day for male, and 18 mg/die for female from menarche to menopause.
Obviously doing so will have to change both the results and the discussion sections, but, in my opinion the use of AR is not suitable for the purpose of assessing the adequacy of any diet, not only for vegetarian ones.
Author Response
REVIEWER 1:
I agree for the changes made to the manuscript and related to other criticisms but I disagree on iron requirements. The AR (average requirements) is defined as the iron intake that cover the requirements of half the population, but this imply that the other half of the population has an inadequate supply of the mineral. Authors should have used the PRI (population reference intake), which is defined as the nutrient intake that cover the requirement of the 97.5% of the population. Using this approach the iron requirement would became 12 mg/day for male, and 18 mg/day for female from menarche to menopause. Obviously doing so will have to change both the results and the discussion sections, but, in my opinion the use of AR is not suitable for the purpose of assessing the adequacy of any diet, not only for vegetarian ones.
Indeed, the AR only covers the requirement of half the population, however this reference value is generally used to assess the risk of inadequacy of nutrition intake in the population, as also recommended by the European Food Safety Authority (EFSA) (EFSA Panel on Dietetic Products, Nutrition and Allergies, 2010). The authors acknowledge that the PRI could be the starting point for the dietary planning for groups, and therefore in the revised version of the manuscript they used PRI to evaluate the nutrient inadequacy of the proposed improved diets with meat replacers (Supplementary Table 4 and Supplementary Table 6). The authors also amended their results and discussion accordingly. Using the population reference intake has an impact on the prevalence of iron inadequacy, in particular prevalence of iron inadequacy is above 50% for women in all countries.
Reference
EFSA Panel on Dietetic Products, Nutrition, and Allergies (NDA); Scientific Opinion on principles for deriving and applying Dietary Reference Values. EFSA Journal 2010; 8(3):1458. [30 pp.]. doi:10.2903/j.efsa.2010.1458. Available online: www.efsa.europa.eu
Reviewer 2 Report
The authors reviseed some contents to reflect the reviewer's comments and others were described as the limitation of study in result section by avoiding the further work to regret.
To clarifying the scope and limitation of study, I would like to suggest that those contents described in results would be moved forward after the purpose of study.
Author Response
REVIEWER 2:
The authors revised some contents to reflect the reviewer's comments and others were described as the limitation of study in result section by avoiding the further work to regret.
To clarifying the scope and limitation of study, I would like to suggest that those contents described in results would be moved forward after the purpose of study.
To the best of our knowledge and expertise, we have addressed the reviewer’s comments. As suggested, in order to further clarify the scope and the limitations of the study, we added the limitation of data variability/uncertainty in the environmental impact estimates of meat replacers to the introduction (lines 68-70 of the revised manuscript). Also, we added the key assumption of using DEA to model dietary improvements, i.e. relying on peer resemblance, to the purpose section of the introduction (lines 72-75 and lines 82-84 of the revised manuscript).
Round 2
Reviewer 1 Report
I have appreciated your discussion of nutrients fullfilment, by the 3 different diets. I agree with your adoption of PRI from EFSA.
In my opinion the manuscript can now be accepted.
This manuscript is a resubmission of an earlier submission. The following is a list of the peer review reports and author responses from that submission.
Round 1
Reviewer 1 Report
It is well known that the greenhhouse effect is reduced when the meat is replaced with plant or plant-based one likely meat replacer, as the gas emission from animal is much greater than plants and the diet becomes healthier. Thus, The first conclusion is clear even without the output obtained from the model applied.
The 100% replacement of meat is only considered the extreme and unrealistic situation. Is it possible to replace the meat to 100%? To evaluate the effect of meat replacer on health and environment, it is more practical that the model should be focused on how the improvement is affected by the replacement ratio.
Also, other factors such as economical impact, cost of processing technology development, etc. crelated to meat replacement were not considered to evaluate the effect on healthy and environmentally sustainable dier. Thus, it seems to be impractical and exaggerated to concluded the efficency.
If the meat diet is fortified, the diet would be healthier. Is any reason not including this condition among the diet model?
Reviewer 2 Report
OVERALL EVALUATION
The role of dietary choice in reducing climate change is the subject of much discussion, so this manuscript is very interesting and covers an innovative topic.
In my opinion there are some points that need to be improved before the paper could be accepted.
MAJOR REMARKS
- Authors have not explained if the environmental impact of mineral fortification has been calculated. If not, it should be calculated, otherwise the calculations is distorted.
- I am aware that data on the concentration of amino acids in food are limited, but when comparing the nutritional value of meat with that of meat-replacer, it is not enough to consider only proteins, but it is also necessary to consider essential amino acids.
- Supplemental table 4. I was surprise by the absence of dietary inadeguacies for iron intakes in Italian women, because the Italian National survey has highlighted an insufficient iron intake in women. This topic need to be clarified.
MINOR REMARKS
Figure 1 is not easy to understand. Authors expressed GHGE as % of CO2 emission of observed diet, but they did not report the global CO2 emission of observed diet.
Line 227 and following. The reported percentage of animal food are refered to the percentage of energy from food of animal origin or have been calculated in another way ?
Reviewer 3 Report
The paper presents the results of a modelling study on the potential role of meat replacers for increased nutritional quality and reduced environmental impact in terms of GHG emissions of food consumption in four European countries (Czech Republic, Denmark, Italy, France). The paper brings novel findings that show the potential for reduced GHG emissions while improving the nutritional quality of the respective diets. Despite the originality and good quality presentation of the results, I recommend making a number of adjustments before the paper gets accepted for publication. I would particularly recommend being more careful with terminology (such as “environmental sustainability”), making a better link between the claims and the data, and extending the limitations of the study as detailed below.
Title:
Please consider adjusting the title – it would be more precise in my view to use “Potential impact of meat replacers on nutritional and environmental indicators of diets in four European countries”
- as far as I understand, the NRD15.3 assesses the nutritional quality of a diet rather than “health” of the population?
- explicitly mentioning that the study dealt with diets in four European countries (rather than diets in the entire EU/Europe) would be useful
Abstract:
Line 30 (and throughout the paper): I strongly disagree with using the term “environmental sustainability” (in Abstract as well as the rest of the paper) while focusing on one environmental indicator only (GHG emissions), numerous publications have shown that a number of environmental aspects need to be considered in order to identify a “safe” or “sustainable” operating space in terms of environmental performance (e.g. the Planetary Boundaries concept by Rockström et al., 2009 published in Nature and followed by Steffen et al., 2015 published in Science). In addition, the potential for reduction of GHG emissions expressed as percentage of the current levels does not say anything about the long-term sustainability, because it does not compare the observed and calculated levels to any benchmark (e.g. the levels of GHG emissions identified to be safe in the PB concept).
I therefore suggest to replace the term “environmental sustainability” by “environmental performance” or “environmental impact” or explicitly “GHG emissions” where relevant
This is the case throughout the paper – e.g. Lines 30, 127, 159, 166, 176, 222, 280, etc.
You even mention in the Discussion (Lines 341-343) that the diets “provide more likely and realistic improvements into the proper direction, which can be gradually progressed, and not yet the ultimate goals for healthy and environmentally sustainable diet…”
If you’d like to include a note on environmental sustainability of the diets in terms of GHG emissions in the paper, then I’d recommend to take the levels of GHG emissions that are considered to be safe in the long term and comparing your values with the published values (for this see e.g. the paper Options for keeping the food system within environmental limits by Springmann et al., 2018 published in Nature and the references therein, among other).
Keywords:
Please consider adding “environmental impact” and/or “GHG emission reduction” (for the moment only “sustainability” is contained on behalf of the environmental dimension of the paper, which is in my view insufficient since the paper primarily deals with GHG emissions as an indicator of environmental impact and not “sustainability” in general, as per my comment on indicators of “sustainability” above)
Introduction:
Line 57: “…are available in supermarkets.” Why the term “supermarkets” only? There are meat replacers available also at farmers markets and small food shops in Europe… I would therefore recommend to use “on the market” or “in retail” or similar.
Line 58: One word likely missing in the sentence „With total and saturated fatty acids….“ – should there be „With total FAT and ….“? Please double check and correct the sentence.
Materials and Methods:
Line 95: Meat-replacement diets
Did you assume the same environmental impact (i.e. GHG emissions) for fortified and unfortified meat replacers? (mostly relates to Line 108-111) This is not mentioned in the methods, but it seems to be the case. It is of course hard to calculate the GHG emissions associated with the additional nutrients and vitamins, but presumably the fortified meat replacers would in effect have a higher environmental impact (GHG emissions) than the unfortified replacers. This should be at least brifely mentioned/discussed in the paper (e.g. in the limitations of the study).
Line 161: Word missing in the following sentence? “The population prevalence of inadequate of the different diets…” Please double check and correct the sentence.
Results:
Line 166 & 176-187 = Figure 1
Where can the reader find the absolute values of GHGE and NRD15.3 in the paper? There are only relative values in % in the results and I could find these values separated for men and women in Supplementary Table 5, rather than seeing the values for the entire population of the countries as presented in Figure 1. This might be useful for comparison with other studies.
Could you also comment on why there seems to be such a big difference in the results of the modelled Max diets between the Czech Republic and the three other countries? I don’t seem to see this in the discussion part of the paper and would be intrigued about your explanation.
Line 193 & 201: The shortcuts “IT”, “FR”, “DK” and “CZ should be explained in methods (so that also non-European readers understand quickly)
Line 206-225 = Figure 2
Are these figures representing results with unfortified meat replacers? This is not clear from the figure caption and the associated text. Please clarify.
Some of the numbers in the figures are illegible – especially in the red part of the figures (meat). Please double check this and adjust.
Line 239-240: “…the amount of fish was more than doubled in all the MaxH diets…” Since fish and eggs are together in Figure 2, this statement is not visible to the readers. Please refer to the relevant table in the Supplementary Material so that the reader can see the data on which this statement is made.
Line 242-243: “…France, with for these population groups the highest amount in the MaxH and MaxS diets…”. There is something missing in this sentence, please double check and adjust to make it meaningful.
Line 250-251: “The total amount of sweet and alcoholic beverages was the lowest in the MaxS diets, while the amount of sweet beverages was the lowest in the MaxH diets.” Since sweet and alcoholic beverages are together in the graph, this is not visible to the readers. Please again refer to the relevant Supplementary Material where the readers can find the data for this.
Line 251-254: Here the text refers to fortified meat replacers, but where can the readers see the data for this? In the Supplementary Material?
Line 256: Supplementary Table 4 – What are the units for results presented in Table 4 (as well as Table 6)? This is not clear to the reader – there are only values without units. Please make a note/explain.
Discussion:
Line 280-295: “In all optimised modelled dies the total amount of meat was lower than in the observed diets, i.e. 30% lower in the MaxP, 50% lower in the MaxH, and 75% lower in the MaxS diets.” Are these numbers average values of all the diets from the four countries?? Same for all the numbers that follow in this paragraph… You presented the results per country in the Results, but here you seem to discuss the average values for all the four countries together? Is this right? Please clarify.
Line 326-327: “Replacing all meat by a (fortified) meat replacer can lead to a 20-30% lower GHGE than observed (Supplemetary Table 5).” There are only absolute values in Suppl. Table 5 – it would be useful for the reader to see both the absolute and the relative values (in %) in this table.
Line 330: Typo “It another modelling study…” should be “In…”
Line 365: Another limitation of the study is the data variability/uncertainty for the GHG emissions of the meat replacers (the values will differ depending on the country of agricultural production of the crops used for the replacers, as well as the country of processing due to e.g. the different energy mix in different countries, due to transport distances etc.). This limitation is currently lacking and must be mentioned in the paper.